# Local Inhibition of Indoleamine 2,3-Dioxygenase Mitigates Renal Fibrosis

**DOI:** 10.3390/biomedicines9080856

**Published:** 2021-07-22

**Authors:** Camilla Grønkjær Jensen, Michael Schou Jensen, Stine Julie Tingskov, Peter Olinga, Rikke Nørregaard, Henricus A. M. Mutsaers

**Affiliations:** 1Department of Clinical Medicine, Aarhus University, 8200 Aarhus, Denmark; cgj@biomed.au.dk (C.G.J.); msj@clin.au.dk (M.S.J.); sjt@clin.au.dk (S.J.T.); rn@clin.au.dk (R.N.); 2Department of Pharmaceutical Technology and Biopharmacy, University of Groningen, 9713 AV Groningen, The Netherlands; p.olinga@rug.nl

**Keywords:** indoleamine 2,3-dioxygenase, 1-methyl-tryptophan, BMS-98620, precision-cut kidney slices, renal fibrosis

## Abstract

Chronic kidney disease (CKD) is a major global health concern and renal fibrosis is an integral part of the pathophysiological mechanism underlying disease progression. In CKD patients, the majority of metabolic pathways are in disarray and perturbations in enzyme activity most likely contribute to the wide variety of comorbidities observed in these patients. To illustrate, catabolism of tryptophan by indoleamine 2,3-dioxygenase (IDO) gives rise to numerous biologically active metabolites implicated in CKD progression. Here, we evaluated the effect of antagonizing IDO on renal fibrogenesis. To this end, we antagonized IDO using 1-methyl-D-tryptophan (1-MT) and BMS-98620 in TGF-β-treated murine precision-cut kidney slices (mPCKS) and in mice subjected to unilateral ureteral obstruction (UUO). The fibrotic response was evaluated on both the gene and protein level using qPCR and western blotting. Our results demonstrated that treatment with 1-MT or BMS-985205 markedly reduced TGF-β-mediated fibrosis in mPCKS, as seen by a decreased expression of collagen type 1, fibronectin, and α-smooth muscle actin. Moreover, IDO protein expression clearly increased following UUO, however, treatment of UUO mice with either 1-MT or BMS-986205 did not significantly affect the gene and protein expression of the tested fibrosis markers. However, both inhibitors significantly reduced the renal deposition of collagen in UUO mice as shown by Sirius red and trichrome staining. In conclusion, this study demonstrates that IDO antagonism effectively mitigates fibrogenesis in mPCKS and reduces renal collagen accumulation in UUO mice. These findings warrant further research into the clinical application of IDO inhibitors for the treatment of renal fibrosis.

## 1. Introduction

Renal fibrosis plays a pivotal role in the development and perpetuation of chronic kidney disease (CKD), which affects approximately 10% of the population [1]. Fibrosis is the end result of a complex cascade of cellular and molecular responses initiated by organ damage, resulting in the loss of functional tissue, and it is regarded as the most damaging process in CKD [2]. Therefore, it is of the utmost importance to identify effective anti-fibrotics that can be used to mitigate CKD progression.

CKD is characterized by numerous changes in enzymatic and metabolic activity, with potential deleterious effects. One of the enzymes that shows altered activity in CKD patients is indoleamine 2,3-dioxygenase (IDO) [3,4]. IDO accounts for 99% of non-protein metabolism of tryptophan, resulting in the formation of numerous biological active metabolites that play an important role in a wide variety of essential biological processes including immune tolerance, antioxidant status, and cell proliferation [5]. Furthermore, it is known that several tryptophan derivates can reduce scar formation [6,7] and suppress TGF-β signaling, which is a key factor in fibrogenesis [8]. On the other hand, IDO expression has been associated with chronic renal failure and can promote renal ischemia-reperfusion injury [9,10]. Additionally, it has been shown that IDO activity is systemically increased in CKD patients [3,4] as well as in unilateral ureteral obstructed (UUO) rats and TGF-β-treated Madin-Darby canine kidney epithelial cells [11]. Moreover, we have previously demonstrated that the plasma levels of kynurenine and quinolinic acid are increased in CKD patients, while kynurenic acid levels are lower in these patients compared to healthy individuals [12]. In addition, it has been reported that the production of melatonin is decreased in CKD patients [13]. These observations further support the notion that CKD patients are in a pro-fibrotic state (elevated kynurenine and quinolinic acid, decreased kynurenic acid and melatonin). Furthermore, it has been demonstrated that several tryptophan metabolites can interfere with the normal physiological function of renal proximal tubule epithelial cells [14,15]. The role of IDO in renal fibrosis has not been widely studied, and it is still not clear whether IDO mitigates or promotes fibrogenesis.

In this study, we investigated the impact of 1-methyl-D-tryptophan (1-MT; also known as Indoximod) and BMS-986205 (also known as Linrodostat) on renal fibrosis. Both are IDO inhibitors, but the mechanism of action is different; 1-MT acts as a potent tryptophan mimetic at the level of IDO effectors [16], while BMS-986205 is a direct suicide inhibitor [16]. The impact of local IDO antagonism on renal fibrosis will be studied in murine precision-cut kidney slices (mPCKS), and the influence of systemic IDO inhibition on the fibrotic process will be investigated in UUO mice.

## 2. Materials and Methods

### 2.1. Ethics Statement

All animal experiments were performed according to the Danish National Guidelines for Animal Care, and were approved by the Danish Veterinary and Food Administration (Approval no. 2015-15-0201-00658).

### 2.2. Experimental Animals

Experiments were performed using male C57BL/6 mice, 7 weeks of age and weighing 19–23 g (Janvier Labs, Le Genest-Saint-Isle, France). All animals had ad libitum access to standard rodent chow (Altromin, Lage, Germany) and tap water. During the experiments, mice were housed in groups of 4–5 mice/cage in a 12 h:12 h light–dark cycle at a temperature of 21 ± 2 °C and a humidity of 55 ± 5%. The animals were allowed to acclimatize to their cages one week prior to surgery.

### 2.3. Experimental Design and Surgical Procedures

During surgery, mice were anesthetized with sevoflurane and placed on a heating pad to maintain an appropriate body temperature (37–38 °C). Through a midline abdominal incision, the left ureter was exposed and occluded with a 6–0 silk ligature. UUO was maintained for seven days. To study the impact of 1-Methyl-D-tryptophan (Sigma Aldrich, St. Louis, MO, USA), 35 mice were randomly divided into four experimental groups; Sham (n = 8), Sham + 10 mg/mL 1-MT (n = 6), 7dUUO (n = 10), and 7dUUO + 10 mg/mL 1-MT (n = 11). To study the effect of BMS-986205 (SMS-Gruppen, Rungsted, Denmark), 22 mice were randomly divided into four experimental groups; Sham (n = 5), Sham + 10 mg/mL BMS-986205 (n = 5), 7dUUO (n = 4), and 7dUUO + 10 mg/mL BMS-986205 (n = 8). 1-MT (diluted in PBS) or BMS-986205 (diluted in DMSO/Corn oil) was administered twice daily via intraperitoneal injection starting at the day of the surgery; control mice were injected with the corresponding solvent control. Dosing was based on previous dose-finding studies performed in our lab, using the following doses of 2, 5, and 10 mg/mL 1-MT, and 10 and 20 mg/mL BMS-986205. After seven days, the kidneys were extracted and blood was collected via cardiac puncture for further analysis. Biochemical analysis of blood samples was performed using a Roche Cobas 6000 analyzer (Roche Diagnostic, Hvidovre, Denmark)) and creatinine levels were determined using the Creatinine Assay Kit (Sigma Aldrich, St. Louis, MO, USA), according to the manufacturer’s instructions.

### 2.4. Precision-Cut Kidney Slices (PCKS)

Mouse PCKS were prepared using a Krumdieck tissue slicer (Alabama Research & Development, Munford, TN, USA) as previously described [17]. PCKS were prepared in ice-cold Krebs-Henseleit buffer, supplemented with 25 mM D-glucose, 25 mM NaHCO_3_, 10 mM HEPES, and saturated with carbogen (95% O_2_, 5% CO_2_) using a Krumdieck tissue slicer. Subsequently, PCKS were cultured in Williams’ Medium E with GlutaMAX containing 10 mg/mL ciprofloxacin and 2.7 g/l D-(+)-Glucose solution at 37 °C in an 80% O_2_, 5% CO_2_ atmosphere while gently shaken. Medium was refreshed every 24 h. PCKS viability was assessed by determining the ATP content of the slices using the ATP Colorimetric/Fluorometric Assay Kit (Sigma Aldrich, St. Louis, MO, USA), according to the manufacturer’s instructions.

### 2.5. Real-Time Quantitative PCR

RNA was isolated using the NucleoSpin RNA Kit (Macherey-Nagel, Düren, Germany), following the manufacturer’s instructions. RNA content was quantified by spectrophotometry and stored at −80 °C until cDNA synthesis. cDNA was synthesized from 0.5 μg RNA with the RevertAid First Strand Synthesis Kit (Thermo Scientific, Roskilde, Denmark). qPCR was performed using the SYBR Green qPCR Master Mix and run on an Aria MX3000p (Agilent Technologies, Lexington, MA, USA). The expression level of the genes of interest was corrected using the reference gene GAPDH or 18S. Primers are listed in Table 1.

### 2.6. Western Blotting

Total protein was extracted using RIPA buffer supplemented with phosphatase-inhibitor 2 and 3 and a mini protease inhibitor tablet. Afterward, 2% SDS and DTT were added to the samples, and they were heated for 15 min at 65 °C. Total protein was separated by SDS/PAGE using 12% Criterion TGX Stain-free gels and subsequently blotted onto a nitrocellulose membrane. Afterward, the membrane was blocked for 1 h with 5% skimmed milk in PBS-T. The blot was then incubated overnight at 4 °C with specific primary antibodies (Table 2). Afterward, the membrane was washed with PBS-T and incubated with the appropriate horseradish peroxidase-conjugated secondary antibody for 1 h at RT. Proteins levels were visualized with ECL-prime detection reagent (Cytiva, Bronshoj, Denmark) and normalized to total protein, as measured using stain-free technology [18].

### 2.7. Histology

Kidneys were fixed by perfusion through the left ventricle using 4% paraformaldehyde (PFA) in water. Afterward, kidneys were immersed in 4% PFA for 1 h, rinsed with PBS, dehydrated using a series of graded alcohol and embedded in paraffin. Subsequently, tissue sections (2 μM) were stained with either Masson’s Trichrome or Sirius red. Next, 8 to 10 pictures were captured in a blinded manner from each sample at 20× magnification with no overlapping regions using an Olympus BX50 light microscope. Severity of fibrosis was quantified with ImageJ 1.52i (http://rsbweb.nih.gov/ij/, accessed on 15 June 2021).

### 2.8. Statistics

Statistics were performed with Graphpad Prism 8.0 (Graphpad Software, Inc., San Diego, CA, USA) via either two-way or one-way ANOVA followed by Tukey’s multiple comparisons test as appropriate. Differences between groups were considered to be statistically significant when *p* < 0.05.

## 3. Results

### 3.1. 1-MT Attenuates TGF-β-Induced Renal Fibrosis in mPCKS

As shown in Figure 1A, treatment with 10 ng/mL TGF-β for 48 h induced fibrogenesis in mPCKS, as seen by the significant increase in the gene expression of the fibrosis markers COL1, FN, and αSMA. Moreover, treatment with 1 mM 1-MT significantly reduced TGF-β-induced gene expression of FN and αSMA (Figure 1A) without affecting viability of the slices (Figure 1D). These results suggest that IDO inhibition can potentially mitigate renal fibrosis. However, even though 1-MT is commonly used as a probe of the IDO pathway, it is not a valid inhibitor of IDO enzyme activity [16]. Thus, to unveil whether IDO antagonism can attenuate renal fibrosis, we tested the antifibrotic efficacy of BMS-986205, a highly potent irreversible IDO inhibitor [16].

### 3.2. BMS-986205 Mitigates TGF-β-Induced Fibrogenesis but Not Inflammation in mPCKS

Our results demonstrated that treatment with BMS-986205 diminishes TGF-β-induced gene expression of COL1, FN, and αSMA (Figure 1A). Moreover, we observed a marked reduction in TGF-β-induced FN and αSMA protein expression following BMS-986205 treatment (Figure 1B) without affecting the viability of the slices (Figure 1D). Conversely, we did not observe any effect of BMS-986205 treatment on TNFα and IL-1β gene expression (Figure 1C). With regard to pro-fibrotic signaling pathways, we did not observe any changes in the gene level of plasminogen activator inhibitor-1 (PAI-1) nor in the pSmad2/Smad2 ratio (Figure 2). Thus, in murine PCKS, inhibition of IDO appears to attenuate renal fibrogenesis without affecting canonical TGF-β signaling.

### 3.3. 1-MT and BMS-986205 Treatment Attenuates Renal Collagen Deposition in UUO Mice

Next, we investigated the antifibrotic efficacy of 1-MT and BMS-986205 in vivo. As shown in Figure 3, UUO markedly increased renal IDO protein expression, supporting the notion that IDO is a suitable therapeutic target. Following surgery, we observed an increase in kidney weight and blood urea nitrogen, indicating the presence of renal injury (Table 3 and Table 4). Administration of 1-MT or BMS-986205 did not affect the weight of the obstructed kidney. In addition, creatinine, plasma sodium, and plasma potassium did not change between the different groups. Moreover, as shown in Figure 4A and Figure 5A, UUO significantly upregulated the gene expression of COL1, COL3, FN, αSMA, TNFα, and IL-1β, as expected [19]. However, in contrast to the results obtained in mPCKS, treatment with either 1-MT or BMS-986205 did not reduce the UUO-induced changes in gene expression (Figure 4A and Figure 5A) with the exception of COL3, of which mRNA levels were significantly diminished following 1-MT treatment (Figure 4A). In addition, western blotting also did not reveal any beneficial effects of 1-MT and BMS-986205 on UUO-induced FN and αSMA protein expression (Figure 4B and Figure 5B). Conversely, treatment with 1-MT and BMS-986205 reduced UUO-induced collagen deposition as shown by Sirius red and trichrome staining (Figure 6). Taken together, these data suggest that IDO antagonism effectively mitigates collagen accumulation, which is the key pathological characteristic of fibrosis, however, the mechanism of action does not appear to revolve around myofibroblast activation nor inflammation.

## 4. Discussion

Renal fibrosis is characterized by the excessive production and accumulation of extracellular matrix proteins such as collagen I, III, VI, and fibronectin, with a detrimental impact on organ architecture and function. Current therapies for renal fibrosis mainly focus on the etiology of CKD such as hypertension or diabetes, and as such, show only limited efficacy in halting the fibrotic process and the concurrent loss of kidney function. Therefore, it is of the utmost importance to find clinically-relevant and druggable targets for the treatment of renal fibrosis. Here, we studied the anti-fibrotic efficacy of 1-MT and BMS-986205, two IDO antagonists, using murine PCKS and UUO mice.

Our results demonstrated that IDO protein expression was markedly increased in UUO mice. This observation is in line with a previous study by Matheus et al., who showed that IDO expression and activity increased in the renal tissue of rats subject to seven days of UUO [11]. Moreover, we and others have reported that IDO activity is elevated in CKD patients, and correlated with disease severity [3,4,20,21,22,23]. This suggests a clinically significant role of tryptophan metabolism in CKD. In contrast, Cheng and colleagues recently set out to evaluate the existence of a causal relationship between metabolites of the tryptophan pathway and kidney function using a bidirectional Mendelian randomization analysis, and they revealed that the increased blood levels of tryptophan metabolites were a consequence rather than a cause of a reduced eGFR [24]. However, it is becoming clear that IDO has a complex role in chronic inflammatory and metabolic diseases and remains an interesting therapeutic target [25], however, in-depth knowledge about its regulation and interconnection to other enzyme systems is essential in the development of effective therapies [26].

Our study is, to the best of our knowledge, the first to demonstrate that local IDO antagonism reduces renal fibrosis. In contrast, Matheus et al. reported that treatment with 1-MT enhanced the pro-fibrotic effect of TGF-β in MDCK cells [11]. This duality is also seen in studies focusing on liver fibrosis. Coa and coworkers reported that IDO inhibition via Danshensu—a bioactive component isolated from Danshen—reduced liver fibrosis in CCl4-treated rats and hepatic stellate cells exposed to TGF-β [27]. In addition, Zhong et al. demonstrated that IDO1 knockout mice were protected from CCl4-induced liver fibrosis [28], and Hoshi et al. reported that knockout of IDO2 as well as treatment with 1-MT attenuated CCl4-induced liver injury [29]. On the other hand, Ogiso and colleagues reported that IDO deficiency aggravated liver fibrosis in CCl4-treated mice [30], and another study reported that IDO inhibition mitigated the anti-fibrotic effect of mesenchymal stem cells in CCl4-treated mice [31]. With regard to the heart, it was reported that co-treatment with IFN-γ and 1-MT can ameliorate cardiac fibrosis in vitro [32]. Taken together, it is clear that more research is needed to fully unravel the role of IDO in organ fibrosis as well as the potential therapeutic effect of IDO inhibition.

In our study, 1-MT and BMS-986205 markedly reduced TGF-β-induced fibrosis in mPCKS without affecting canonical TGF-β signaling in the case of BMS-986205, while in UUO mice, we only observed a reduction in collagen deposition. These results underscore the complex role of IDO in fibrosis. Moreover, it suggests that targeted IDO antagonism might be more beneficial as compared to systemic treatment with IDO inhibitors. To this end, one could improve the pharmacological properties of both compounds by using a drug-targeting strategy that is specifically directed to activated fibroblasts using the platelet-derived growth factor receptor beta (PDGFRβ) as a docking receptor. Using this approach, Poosti et al. successfully prepared an IFNγ conjugate targeted to PDGFRβ that mitigated fibrogenesis in murine PCKS [33]. In another study, it was demonstrated that sustained systemic delivery of a similar anti-fibrotic construct could be achieved following a single subcutaneous injection of construct-loaded polymeric microspheres [34]. These exciting advances in drug formulation will allow us to directly target complex systems such as tryptophan metabolism to treat disease without causing severe side effects.

In conclusion, this study demonstrates that IDO antagonism effectively mitigates fibrogenesis in mPCKS and reduces renal collagen accumulation in UUO mice. These findings warrant further research into the clinical application of IDO inhibitors for the treatment of renal fibrosis.

## Figures and Tables

**Figure 1 biomedicines-09-00856-f001:**
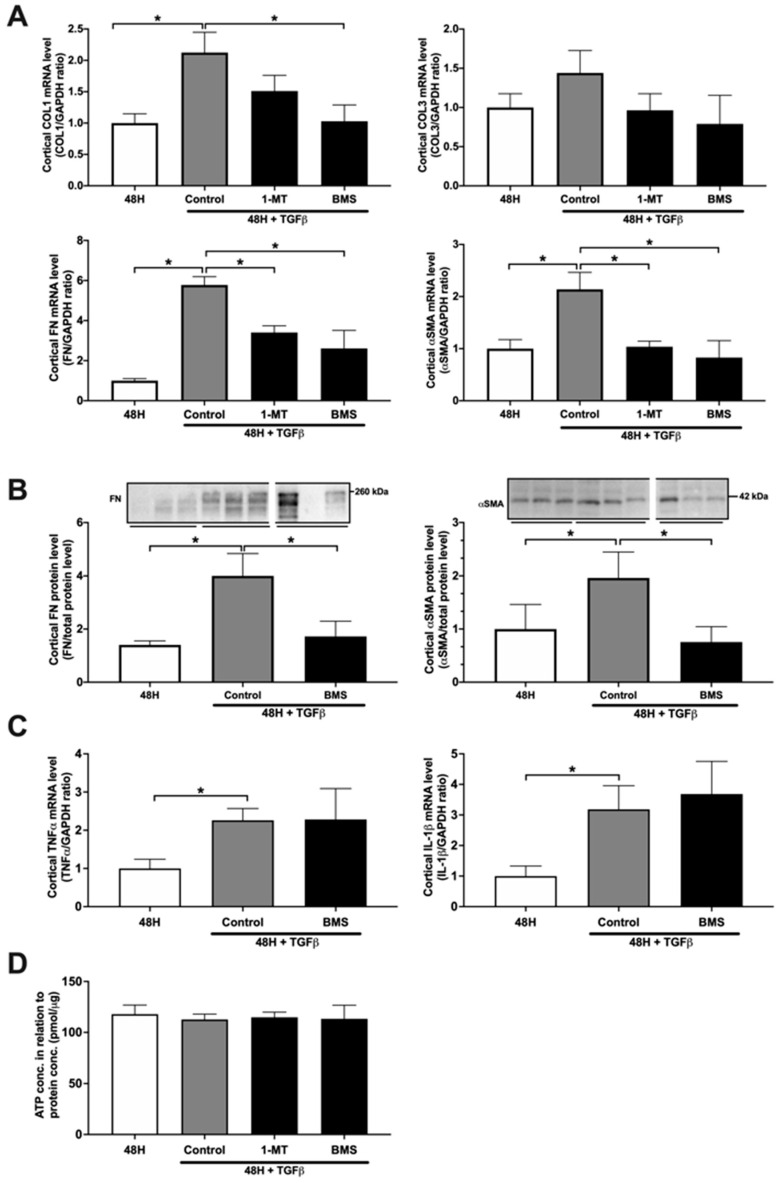
1-methyl-D-tryptophan and BMS-986205 mitigate TGF-β-induced fibrosis in murine PCKS. PCKS were exposed to 10 ng/mL TGF-β in the absence or presence of 1-methyl-D-tryptophan (1 mM; 1-MT) or BMS-986205 (5 nM) for 48 h. (**A**) Relative mRNA expression of the fibrosis markers collagen 1A1 (COL1), collagen 3A1 (COL3), fibronectin (FN), and α-smooth muscle actin (αSMA) normalized to GAPDH. n = 4–10. (**B**) Western blotting was used to study the protein expression of fibronectin (FN) and α-smooth muscle actin (αSMA). n = 7. (**C**) Relative mRNA expression of the inflammation markers tumor necrosis factor alpha (TNFα) and interleukin 1 beta (IL-1β) normalized to GAPDH. n = 6–7. (**D**) Viability of mPCKS after treatment, assessed by ATP content of the slices. n = 4–10. Data are presented as mean ± SEM. * *p* < 0.05.

**Figure 2 biomedicines-09-00856-f002:**
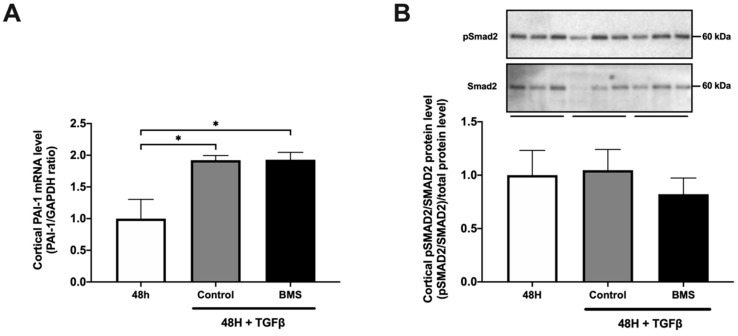
BMS-986205 does not alter canonical TGF-β signaling in murine PCKS. PCKS were exposed to 10 ng/mL TGF-β in the absence or presence of BMS-986205 (5 nM) for 48 h. (**A**) Relative mRNA expression of plasminogen activator inhibitor-1 (PAI-1) normalized to GAPDH. n = 3–7. (**B**) Western blotting was used to study the protein expression of total and phosphorylated Smad2. n = 6. Data are presented as mean ± SEM. * *p* < 0.05.

**Figure 3 biomedicines-09-00856-f003:**
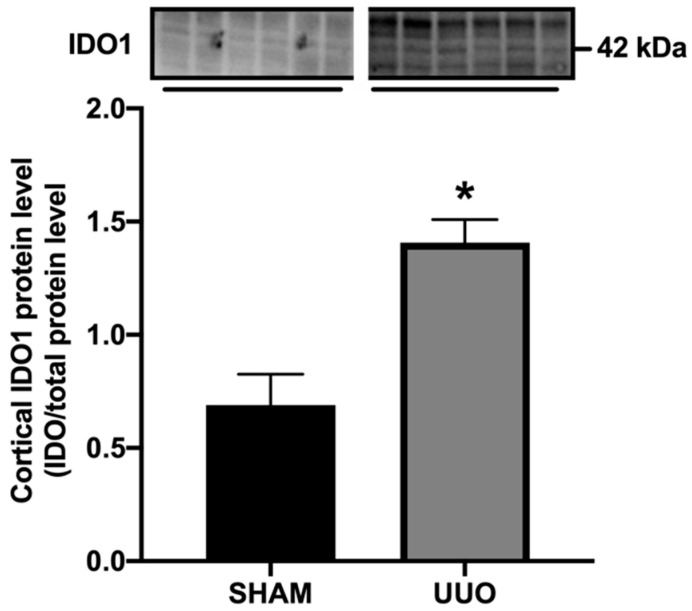
Cortical IDO protein expression increases following UUO. Mice were subjected to 7 days of unilateral ureteral obstruction (UUO). Afterward, indoleamine 2,3-dioxygenase (IDO) protein expression was studied using western blot (n = 8–10). Data are presented as mean ± SEM. * *p* < 0.05.

**Figure 4 biomedicines-09-00856-f004:**
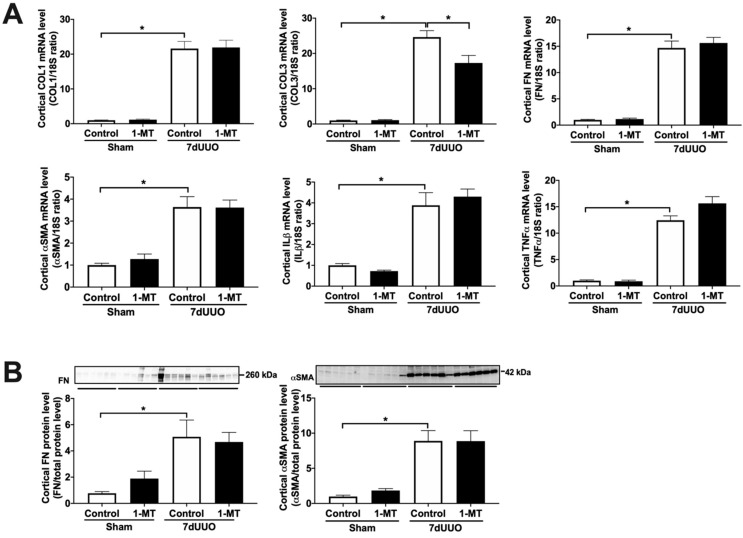
1-Methyl-D-tryptophan solely reduced collagen 3A1 gene expression in UUO mice. Mice were subjected to 7 days of unilateral ureteral obstruction (UUO) and treated with 1-methyl-D-tryptophan (10 mg/mL). (**A**) Relative mRNA expression of the fibrosis markers collagen 1A1 (COL1), collagen 3A1 (COL3), fibronectin (FN), and α-smooth muscle actin (αSMA), and the inflammation markers tumor necrosis factor alpha (TNFα) and interleukin 1 beta (IL-1β), all normalized to 18S. n = 6–11. (**B**) Protein expression of fibronectin (FN) and α-smooth muscle actin (αSMA) was studied using western blot, corrected for total protein levels. n = 6–11. Data are presented as mean ± SEM. * *p* < 0.05.

**Figure 5 biomedicines-09-00856-f005:**
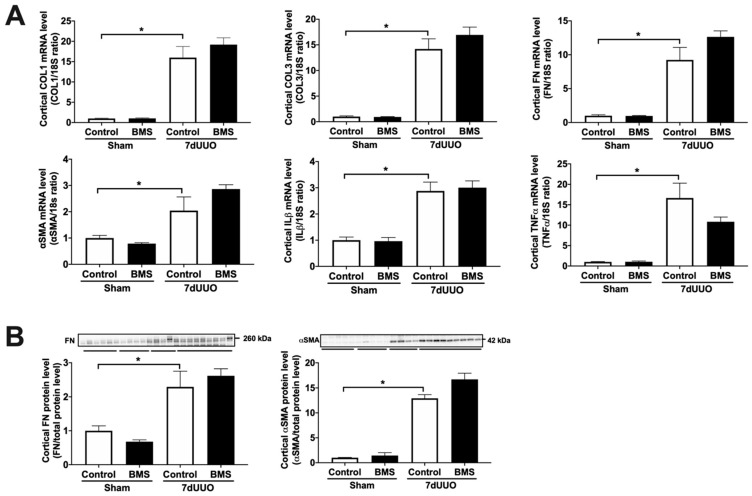
BMS-986205 does not reduce the gene and protein expression of fibrosis markers in UUO mice. Mice were subjected to 7 days of unilateral ureteral obstruction (UUO) and treated with BMS-986205 (10 mg/mL). (**A**) Relative mRNA expression of the fibrosis markers collagen 1A1 (COL1), collagen 3A1 (COL3), fibronectin (FN), and α-smooth muscle actin (αSMA), and the inflammation markers tumor necrosis factor alpha (TNFα) and interleukin 1 beta (IL-1β), all normalized to 18S. n = 4–7. (**B**) Protein expression of fibronectin (FN) and α-smooth muscle actin (αSMA) was studied using western blot, corrected for total protein levels. n = 4–9. Data are presented as mean ± SEM. * *p* < 0.05.

**Figure 6 biomedicines-09-00856-f006:**
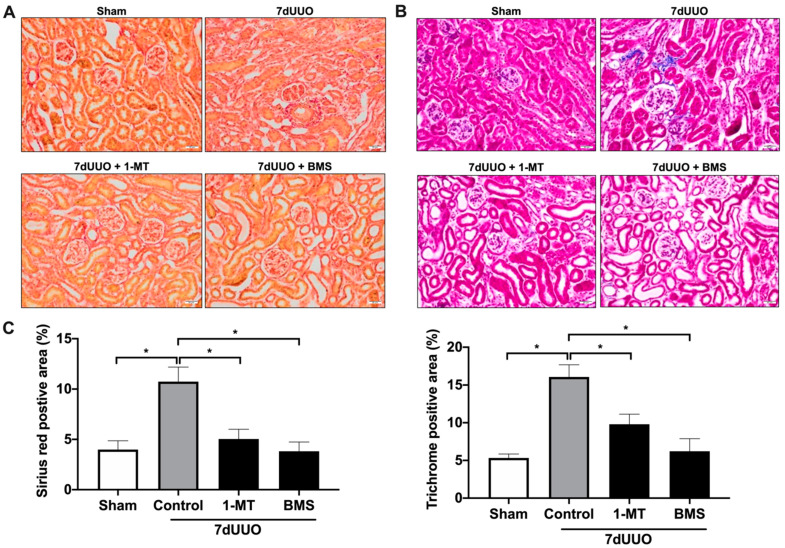
1-Methyl-D-tryptophan and BMS-986205 markedly reduce collagen deposition in UUO mice. Mice were subjected to 7 days of unilateral ureteral obstruction (UUO) and treated with either 1-methyl-D-tryptophan (10 mg/mL) or BMS-986205 (10 mg/mL). (**A**) Representative images of Picrosirius red staining. (**B**) Representative images of Masson’s trichrome staining. (**C**) Quantification of Picrosirius red and Masson’s trichrome staining as a percentage of the total area of the tissue from Sham and UUO mice. Magnification 20×. Scale bar: 20 μm. n = 3–5. Data are presented as mean ± SEM. * *p* < 0.05.

**Table 1 biomedicines-09-00856-t001:** qPCR primers.

*Target Gene*	Forward Primer	Reverse Primer
*COL1*	5′-CACCCTCAAGAGCCTGAGTC-3′	5′-ACTCTCCGCTCTTCCAGTCA-3′
*COL3*	5′-GCACAGCAGTCCAACGTAGA-3′	5′-TCTCCAAATGGGATCTCTGG-3′
*FN*	5′-AATGGAAAAGGGGAATGGAC-3′	5′-CTCGGTTGTCCTTCTTGCTC-3′
*αSMA*	5′-CTGACAGAGGCACCACTGAA-3′	5′-CATCTCCAGAGTCCAGCACA-3′
*TNFα*	5′-AGGCTGCCCCGACTACGT-3′	5′-GACTTTCTCCTGGTATGAGATAGCAAA-3′
*IL-1β*	5′-CAGGCAGGCAGTATCACTCA-3′	5′-TGTCCTCATCCTGGAAGGTC-3′
*PAI-1*	5′-AGTCTTTCCGACCAAGAGCA-3′	5′-GACAAAGGCTGTGGAGGAAG-3′
*GAPDH*	5´-TAAAGGGCATCCTGGGCTACACT-3′	5′-TTACTCCTTGGAGGCCATGTAGG-3′
*18S*	5′-TGTGGTGTTGAGGAAAGCAG-3′	5′-TCCCATCCTTCACATCCTTC-3′

**Table 2 biomedicines-09-00856-t002:** Primary antibodies used for western blot.

*Target*	Catalog No.	Company	Dilution
FN	Ab2413	Abcam	1:2000
αSMA	MO0851	Dako	1:500
Smad2	3102	Cell signaling	1:2000
pSmad2	3108	Cell signaling	1:500
IDO1	MAB5412	Merck Millipore	1:500

**Table 3 biomedicines-09-00856-t003:** Functional data after UUO and 1-MT treatment.

	Sham	Sham + 1-MT	7dUUO	7dUUO + 1-MT
Bodyweight (g)	21.7 ± 1.4	22.5 ± 1.9	21.8 ± 1.9	22.2 ± 1.4
Left kidney/Bodyweight (mg/g)	5.8 ± 0.7	5.7 ± 0.3	6.7 ± 0.6 *	6.5 ± 0.3 *
P_Na_ (mmol/L)	146.8 ± 1.28	148 ± 1.58	147.4 ± 1.33	147.4 ± 1.43
P_K_ (mmol/L)	4.5 ± 0.48	4.3 ± 0.14	4.4 ± 0.27	4.5 ± 0.42
Creatinine (mmol/L)	12.6 ± 5.32	18.0 ± 3.97	17.3 ± 3.35 *	16.1 ± 4.44
Osmol (mOsmol/kg)	333.1 ± 18	337.8 ± 8.07	331 ± 9.31	329.3 ± 9.61
BUN (mmol/L)	4.9 ± 1.61	5.5 ± 1.46	6.6 ± 1.33 *	6.5 ± 1 *

Values are presented as mean ± SD. Sham (n = 8), Sham + 1-MT (n = 6), 7dUUO (n = 10), 7dUUO + 1-MT (n = 11). UUO; unilateral ureteral obstruction, P_Na_; plasma sodium, P_K_; plasma potassium, BUN; Blood urea nitrogen. * *p* < 0.05 compared with sham-operated mice.

**Table 4 biomedicines-09-00856-t004:** Functional data after UUO and BMS-986205 treatment.

	Sham	Sham + BMS	7dUUO	7dUUO + BMS
Bodyweight (g)	24.15 ± 1.9	23.15 ± 0.4	21.66 ± 1.3 *	21.92 ± 1.4 *
Left kidney/Bodyweight (mg/g)	5.8 ± 0.7	5.9 ± 0.3	6.4 ± 0.5	6.8 ± 0.8 *
P_Na_ (mmol/L)	150 ± 1.63	149.8 ± 0.84	150.8 ± 0.45	150.4 ± 3.10
P_K_ (mmol/L)	4.4 ± 0.50	4.2 ± 0.24	5.0 ± 0.72	4.7 ± 0.63
Creatinine (mmol/L)	13.3 ± 4.82	10.2 ± 4.05	14.9 ± 3.18	12.1 ± 2.65
Osmol (mOsmol/kg)	313.8 ± 2.22	311 ± 4	321.6 ± 14.77	323.5 ± 14.88
BUN (mmol/L)	5.8 ± 1.02	5.1 ± 1.05	7.3 ± 0.81*	6.3 ± 1.03

Values are presented as mean ± SD. Sham (n = 5), Sham + BMS-986205 (n = 5), 7dUUO (n = 4), 7dUUO + BMS-986205 (n = 8). UUO; unilateral ureteral obstruction, P_Na_; plasma sodium, P_K_; plasma potassium, BUN; Blood urea nitrogen. * *p* < 0.05 compared with sham-operated mice.

## Data Availability

All data generated or analyzed during this study are included in the published article.

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
