# Peer review of "Local Inhibition of Indoleamine 2,3-Dioxygenase Mitigates Renal Fibrosis"

_biomedicines, 2021, doi:10.3390/biomedicines9080856_

Round 1
Reviewer 1 Report
This study is interesting, but seems biochemically disengaged and overly simplistic.
They show, in a model system, that inhibition of IDO reduces renal fibrosis, but do not place this into the context of fibrotic pathology that can, according to diverse previous studies, emerge either in the presence or absence of robust IDO function.
For example, if you block IDO and follow the non-IDO pathway all the way through to the end, you arrive at an anti-fibrotic substance (melatonin). However, if the non-IDO biochemistry is hindered before reaching the melatonin conclusion, you produce residual pro-fibrotic substances (serotonin and its close analogs).
By contrast, if you allow IDO to serve its purpose, it will also eventually lead to an anti-fibrotic substance (kyurenic acid), **but only** if that full pathway is allowed to progress to conclusion, and not leave you with pro-fibrotic substances like L-kynurenine or, especially, N-formyl kyurenine.
So, in a broadly general sense, fibrosis seems to depend less on the rate of IDO-mediated tryptophan, and more on whether the body's physiology can robustly and fully metabolize tryptophan via either an IDO-mediated or non-IDO-mediated route.
In the current study, the authors seem to have found a model system that produces a general reduction in fibrosis by abrogating IDO activity, but I would argue that their model system may be artificially limited to a narrow phenotype where post-IDO metabolite processing is compromised, while serotonin metabolism (i.e., downstream from IDO-blocking) is fully robust. In practice, there may be patients for whom serotonin processing is highly robust while kynurenine processing is faulty... but instinctively I would expect that to be more the exception than the rule.
To fully justify their study and strengthen its impact, I think the authors should provide clear evidence that a significant fraction of renal fibrosis patients do, indeed, have a pronounced imbalance between kynurenine versus serotonin metabolism.
Author Response
We thank the Reviewer for the extremely insightful comment. It is absolutely true that several tryptophan metabolites can have either anti- or pro-fibrotic effects, and it is difficult to fully comprehend how targeting a single enzyme will alter the whole downstream metabolic cascade. Nonetheless, in the current manuscript we study the therapeutic potential of IDO inhibition for the treatment of renal fibrosis in the setting of chronic kidney disease (CKD). We have previously demonstrated that the plasma levels of kynurenine and quinolinic acid are increased in CKD patients, while kynurenic acid levels are lower as compared to healthy individuals.1 In addition, it has been reported that the production of melatonin is decreased in CKD patients.2 These observations are in line with the comment of the Reviewer, and thus, we believe that it can be postulated that CKD patients are in a pro-fibrotic state (elevated kynurenine and quinolinic acid, decreased kynurenic acid and melatonin). Next to this, it has been suggested that – in dendritic cells – IDO is not only an enzyme but can also acts as a signal transducer, promoting the activation of the genes encoding IDO, TGF-β and IFN-α–IFN-β.3 If this holds true for other cells and tissues, it implies that IDO can promote fibrosis by activating pro-fibrotic TGF-β signaling.
Still, we agree with the Reviewer that tryptophan metabolism is complex and that future research should include extensive metabolic profiling. A statement on the imbalance in tryptophan metabolites in CKD patients has been added to the Introduction.
References:
1 van den Brand, J. A. et al. Uremic Solutes in Chronic Kidney Disease and Their Role in Progression. PLoS One 11, e0168117, doi:10.1371/journal.pone.0168117 (2016).
2 Rahman, A., Hasan, A. U. & Kobori, H. Melatonin in chronic kidney disease: a promising chronotherapy targeting the intrarenal renin-angiotensin system. Hypertens Res 42, 920-923, doi:10.1038/s41440-019-0223-9 (2019).
3 Chen, W. IDO: more than an enzyme. Nat Immunol 12, 809-811, doi:10.1038/ni.2088 (2011).
Reviewer 2 Report
The authors show that local inhibition of indoleamine 2, 3 dioxygenase decreases renal fibrosis in mPCKS model. Usually, renal fibrosis is complex with the production of extracellular matrix and infiltration of inflammatory cells and cytokine expression. However, in this in vivo study, by using the UUO model, inhibition of IDO does not show the protective effect on UUO-induced fibrosis, only reduces renal collagen deposition. So, authors need to address more precise mechanisms in ex vivo experiments, which show a decrease of extracellular matrix.
- Western blot should show the more larger and precise figures and add the protein loading control such as β-actin or GAPDH.
- Page 2, line 23: there is a missing spell. Please check.
- How about the expression of the downstream signaling pathway of TGF-β in mPCKS model?
- In figure 5B, treatment of 1-MT in the UUO model group show more collagen and inflammatory cells deposition compared to the 7dUUO kidney. The authors need to re-evaluate the trichrome positive area in Figure 5B.
Author Response
Western blot should show the more larger and precise figures and add the protein loading control such as β-actin or GAPDH.
We have amended the figures to included larger Western blot images as requested by the Reviewer. Protein loading was checked with stain-free technology, which is a relative new technology developed by BIO-RAD Laboratories, and has several advantages; 1) allows to assure equal loading of proteins to the gel, 2) verifies equal transfer of protein from gel to nitrocellulose membrane, and 3) allows normalization to total protein measured on the same membrane used for detection of protein of interest.1,2
When using housekeeping proteins, assumptions of constitutive expression in all samples across all experimental conditions must be made. However, several studies have demonstrated inconsistency in the expression of several housekeeping proteins often used as reference.3 In addition, for accurate quantification, the protein must be loaded within the linear detection range. However, housekeeping proteins are often highly abundant and the risk of overloading is high, whereby regulation between experimental groups may not be detected.4 Housekeeping proteins must be detected on the same membrane to ensure the same accuracy as stain-free technology. This complicates probing both the housekeeping protein and a low abundance target protein on the same membrane. To ensure copious amount of protein to detect the target protein, the housekeeping protein will most likely be outside the linear detection range. In addition, probing both the target protein and the reference is not always possible due to coinciding molecular weight. Finally, the use of housekeeping protein for data normalization is based on a single protein compared with stain-free technology that is based on the total of all proteins in a given sample, which in summary may leave the stain-free technology superior to the use of housekeeping proteins.
Page 2, line 23: there is a missing spell. Please check.
We thank the Reviewer for pointing this out. We have carefully checked and removed spelling mistakes from the manuscript.
How about the expression of the downstream signaling pathway of TGF-β in mPCKS model?
To evaluate the impact of BMS-986205 on TGF-β signaling, we investigated the gene expression of plasminogen activator inhibitor-1 (PAI-1) as well as Smad2 phosphorylation. As shown in the new Figure 2, TGF-β markedly increased the gene expression of PAI-1 and promoted the phosphorylation of Smad2, however, these changes were not significantly affected by treatment with 5 nM BMS-986205. These findings suggest that BMS-986205 possibly elicits its antifibrotic effects by targeting the non-canonical (non-Smad2/3) pathway.5
In figure 5B, treatment of 1-MT in the UUO model group show more collagen and inflammatory cells deposition compared to the 7dUUO kidney. The authors need to re-evaluate the trichrome positive area in Figure 5B.
We agree with the Reviewer that the provided images did not accurately reflect our analysis, therefore we re-evaluated the staining and included better representative images.
References:
1 Gurtler, A. et al. Stain-Free technology as a normalization tool in Western blot analysis. Anal Biochem 433, 105-111, doi:10.1016/j.ab.2012.10.010 (2013).
2 Posch, A., Kohn, J., Oh, K., Hammond, M. & Liu, N. V3 stain-free workflow for a practical, convenient, and reliable total protein loading control in western blotting. J Vis Exp, 50948, doi:10.3791/50948 (2013).
3 Ferguson, R. E. et al. Housekeeping proteins: a preliminary study illustrating some limitations as useful references in protein expression studies. Proteomics 5, 566-571, doi:10.1002/pmic.200400941 (2005).
4 Dittmer, A. & Dittmer, J. Beta-actin is not a reliable loading control in Western blot analysis. Electrophoresis 27, 2844-2845, doi:10.1002/elps.200500785 (2006).
5 Finnson, K. W., Almadani, Y. & Philip, A. Non-canonical (non-SMAD2/3) TGF-beta signaling in fibrosis: Mechanisms and targets. Semin Cell Dev Biol 101, 115-122, doi:10.1016/j.semcdb.2019.11.013 (2020).
Round 2
Reviewer 1 Report
Through revised text, plus clear citation of prior work (including some of their own prior relevant work), the authors have effectively addressed my concerns regarding the significance of kynurenine / kynurenic acid ratio as a plausible pathology marker.
Reviewer 2 Report
This revised manuscript has adequately address my comments.